# Dominant effects of the immediate environment on the gut microbiome of mice used in biomedical research

Aaron C. Ericsson,[1,2,3] Zachary L. McAdams,[1] Rebecca A. Dorfmeyer,[2,3] Marcia L. Hart,[4] Armedia O'Neill-Blair,[2] James Amos-Landgraf,[1,2,3] Craig L. Franklin[1,2,3]

**ABSTRACT** Studies using genetically engineered mouse (GEM) models are often performed over extended periods. The microbiomes of GEM colonies are expected to retain some of the microbial features present in the founder mice used to generate each GEM model and to acquire new features through dietary and environmental sources. The rate at which these processes occur over time likely varies between institutions. To assess the relative effect size of environment on the microbiome of GEMs used in biomedical research, we performed 16S rRNA metabarcoding of fecal samples from 275 distinct GEM lines ($n = 351$) maintained by 139 different laboratories at 84 different research institutions in 34 U.S. states or districts and seven other countries, and compared intra-strain, inter-strain, inter-lab, and inter-institution similarities. Reference data from mice harboring supplier-origin (SO) microbiomes ($n = 1,171$) were used to determine the relative contribution and nature of microbes from known and unknown sources. Paradoxically, the data indicate that the immediate laboratory-level environment is the dominant factor shaping the microbiome of GEM models, but that the microbiome of GEMs develops similarities in beta-diversity, regardless of other factors. Related to this, we detected an unexpectedly high prevalence and abundance of *Helicobacter* spp. in GEM microbiomes, the abundance of which correlated significantly with the abundance of multiple resident taxa colonizing the mucosa. These findings suggest a higher prevalence of *Helicobacter* spp. in laboratory mice than previously appreciated, and the possibility of positive and negative interactions with other taxa is found to affect GEM model phenotypes.

**IMPORTANCE** There are concerns regarding the reproducibility and predictive value of mouse models of human disease. Notwithstanding those legitimate concerns, genetically engineered mouse (GEM) models provide an invaluable platform to investigate gene function or effects of environmental factors in a biological system. The microbiome of GEM models significantly influences model phenotypes and thus represents a possible source of poor reproducibility. While the microbiome is often incorporated in research investigating disease mechanisms using GEMs, limited information is available regarding the similarity of the microbiome of GEM models within and between research labs at the same institution, or across institutions. Moreover, while the microbiome of founder mice from different suppliers is known to differ, the degree to which features present in supplier-origin microbiomes are retained in GEM colonies throughout experimentation is unclear. These data demonstrate the robust effect of lab-level environment and the need for sample collection concurrent with phenotyping.

**KEYWORDS** gut microbiome, mouse, mouse model

**Peer Reviewers** Axel Kornerup Hansen, Kobenhavns Universitet, Frederiksberg C, Denmark; Jeffrey Price, University of Nebraska-Lincoln, Lincoln, Nebraska, USA

Address correspondence to Aaron C. Ericsson, ericssona@missouri.edu.

The authors declare no conflict of interest.

See the funding table on p. 15.

H ost-associated microbiomes are essential components of holobiont fitness. Required for optimal energy harvest and diversification of dietary compounds

(1), the gut microbiome (GM) also induces maturation of the immune and central nervous systems (2, 3) and colonization resistance against pathogenic organisms (4). Each host species has co-evolved with its own native GM (5), comprising highly host-specific bacteria from the same dominant phyla across most terrestrial mammals (6, 7). While most naturally born offspring initially acquire their GM primarily from maternal sources (8, 9), diet and other environmental factors can influence GM composition during development or adulthood. Similarly, the offspring genome, representing a combination of maternal and paternal germlines, can shape the composition of the human and rodent GM (10, 11).

The GM has become a point of interest in mouse models, as a potential source of poor reproducibility, but also as a clue regarding possible disease mechanisms. Considerable efforts have been made to elucidate the effects of intrinsic (e.g., genotype, sex, age) and extrinsic (e.g., antimicrobial exposure, husbandry, diet) factors on the GM of specific pathogen-free (SPF) laboratory mice (11–17). Notably, the commercial supplier of the mouse has been reported by multiple groups (15, 18–22) as a major determinant of the microbiome of mice used in biomedical research. As the GM is transmitted vertically during regular mating, features inherent in the GM of colony founders are presumably maintained across generations, albeit under the influence of diet and other environmental pressures and sources of novel bacteria.

Colonies of outbred CD-1 mice harboring distinct supplier-origin (SO) GMs were established in 2017 (23) and have been maintained continuously since then at the Mutant Mouse Resource and Research Center at the University of Missouri (MU MMRRC). Mice from these colonies are used during embryo transfer (ET) rederivation of other mouse lines to generate one or more isogenic colonies of mice with distinct SO microbiomes (24–27). These and other studies (28–37) show that the differences between SO GMs are clinically relevant in a wide range of phenotypes and disease models. Many of the effects on model phenotype can be traced to specific taxa that vary in presence between SO GMs, such as segmented filamentous bacteria (SFB, *Candidatus Savagella*) and *Mucispirillum* spp. (38, 39). Similarly, the phenotypes of many mouse models of colitis are absolutely dependent on purposeful infection with *Helicobacter* spp. (40–47), which are not present in most mouse production colonies but are commonly detected in conventionally housed mice (48) and in sentinel testing of research colonies (49).

Genetically engineered mouse (GEM) models include global and conditional gene knockout, knock-in, and transgenic strains, as well as an ever-expanding range of phenotypically silent lines, such as Cre-driver strains. These mouse strains are generated on a variety of genetic backgrounds, typically using mice originating from one of four commercial suppliers and maintained under varying conditions at institutions across the world. In the U.S.A., a large number of these mutant mouse lines are maintained as cryopreserved germplasm or embryos at one of the four centers within the NIH-funded Mutant Mouse Resource and Research Center (MMRRC) consortium. Fecal samples from all mutant mouse lines submitted to the MU MMRRC are collected upon arrival and subjected to 16S rRNA amplicon sequencing to document the microbiome prior to cryopreservation. As a collection, these data provide novel information regarding the population-level native microbiome of research mice.

During resuscitation of cryopreserved mouse lines, it may be desirable to do so using ET recipients with a GM similar to that of the mice submitted for cryopreservation. Similarly, should the model phenotype of a resuscitated line differ from published data, archived information on the ancestral GM becomes a valuable reference. Ultimately, these uses require a means of classifying and reporting the native GM prior to cryopreservation. As SO GMs represent dominant determinants of the laboratory mouse microbiome (and significant sources of phenotypic variability), reference data from the four suppliers might serve as standards against which data from cryopreserved lines can be compared.

Here, we characterized the GM of mutant mice (351 mice from 275 distinct lines) submitted to the MU MMRRC and determined the relative similarities among mice at the level of institution, laboratory, and strain. We then used data from mice harboring different SO GMs as reference data to compare beta-diversity and determine the relative contribution of the four domestic SO GMs, as well as unknown sources, to the microbiome of cryopreserved GEMs. Lastly, correlation analyses were performed using data from GEMs to identify cooperative and competitive relationships between taxa that are conserved across a range of genotypes and environments.

## MATERIALS AND METHODS

### Mice – supplier-origin (SO) gut microbiome (GM) controls

Crl:CD1 (CD1, Charles River Laboratories) colonies of mice were established in 2015 via embryo transfer (ET) rederivation of Crl:CD1 germplasm in surrogate dams purchased from the Jackson Laboratory (C57BL/6J), Taconic Biosciences (C57BL/6NTac), Charles River Laboratories (Crl:CD1), or Envigo (now Inotiv, Hsd:ICR). CD1 offspring of dams from each supplier were then used as founders for each colony. These supplier-origin gut microbiomes (GM) harbored by each colony were originally named GM1 through GM4, in order of mean observed richness (23). Table 1 shows the source of each reference GM.

All four colonies were maintained at the MU MMRRC for roughly two years before GM2 and GM3 were discontinued. GM1 and GM4 (renamed $GM^{Low}$ and $GM^{High}$ based on a history of lowest and highest mean richness among suppliers) have been maintained continuously since their establishment, avoiding inbreeding and introducing new genetic stock into each colony via ET rederivation of embryos collected from newly purchased Crl:CD1 mice (Charles River Laboratories) into surrogate dams from the existing colonies. This is performed annually in each colony using a minimum of three embryo donors per colony.

Colonies were kept on separate individually ventilated cage (IVC) racks (Thoren) within the same animal room, maintained between 68–79°F and 30–70% humidity. Cage changes were performed on separate days for each colony, using sterilized instruments within a biosafety cabinet, and all cages are kept under positive pressure. Mice were group-housed with four mice per cage. Mice had *ad libitum* access to autoclaved maintenance chow (LabDiet 5058) and to autoclaved, acidified drinking water. Mice were housed on compressed paper chip bedding (Shepherd Specialty Papers, Watertown, TN) until early 2024. Following the discontinuation of that bedding product by the manufacturer, mice were housed on corncob bedding (Andersons Lab Bedding Products, Maumee, OH). Each cage receives environmental enrichment consisting of one nestlet (Ancare, Bellmore, NY) and one-half portion of crinkle nest (Andersons Lab Bedding Products). Mice were monitored for pathogens with sentinel testing through IDEXX BioAnalytics (Columbia, MO). Quarterly sentinel testing includes serologic testing for MHV, MVM, MPV, MNV, TMEV, EDIM, Sendai virus, *Mycoplasma pulmonis*, PVM, REO3, LCMV, ECTV, MAV1, MAV2, Polyoma virus, and *Pneumocystis murina*; PCR testing for *Helicobacter* spp. (with speciation of positives), *M. pulmonis*, and beta-hemolytic streptococci (Groups A, B, C, G); parasitologic evaluation for fur mites, mesostigmatid mites, lice, and flagellates including *Spironucleus muris*, *Giardia muris*, *Hexamastix muris*, *Trichomonas muris*, *Tritrichomonas muris*, *Entamoeba muris*, pinworms, and tapeworms; and microbiologic evaluation (i.e., traditional culture) for *Citrobacter rodentium*, *Klebsiella oxytoca*, *Klebsiella pneumoniae*, *Rodentibacter pneumotropicus*, *Rodentibacter heylii*, *Streptococcus pneumoniae*, *Salmonella* spp., and *Bordetella hinzii*. Annually, additional colony surveillance is performed, including serologic testing for *Encephalitozoon cuniculi*, *Filobacterium rodentium*, *Clostridium piliforme*, MCMV, K virus, LDEV, Hantaan virus, and MTV; PCR testing for *Cryptosporidium* sp. and *Streptobacillus moniliformis*; and microbiologic evaluation for *Corynebacterium kutscheri*, *Corynebacterium bovis*, *Pasteurella multocida*, and *Bordetella bronchispetica*.

**TABLE 1** List of the four reference gut microbiomes (GM) used throughout these analyses, including the original supplier and mouse strain or stock that provided each GM to offspring via embryo transfer surrogate dams[a]

| Gut microbiome (GM) | Source (supplier) | GM source (strain or stock) |
|---|---|---|
| GM1 (GM[Low]) | The Jackson Laboratory | C57BL/6J |
| GM2 | Taconic Biosciences | C57BL/6NTac |
| GM3 | Charles River Laboratories | Crl:CD1 |
| GM4 (GM[High]) | Envigo (Inotiv) | Hsd:ICR |

[a]Also provided is the mean (± SD) observed richness in these colonies, based on the colony survey data.

Colonies of FVB mice were generated in 2018 via ET rederivation of FVB/NCrl (Charles River Laboratories) mice using the same surrogate dams from each of the four domestic suppliers of research mice that were used to generate the CD1 colonies. Colonies of C57BL/6J (B6J) and C57BL/6J-Apc[Min]/J (the Jackson Laboratory) were generated similarly with only GM1 (GM[Low]) or GM4 (GM[High]).

## Mice – genetically engineered mice (GEM)

Mutant mice included in the study were submitted for cryopreservation at the MU MMRRC by investigators at 84 different research institutions, with representation from North America, Europe, Asia, and Australia. Fresh fecal samples were collected immediately from unmanipulated mice as they were being unpacked from shipping containers to avoid any normalizing effect of our own institution. Prior to submission, GEM mice were subjected to the diet and husbandry conditions of the vivaria in which they were housed at each different institution.

## Sample collection

Fecal samples from the CD1 colonies have been collected on a quarterly basis since 2019. Pooled feces from 10 cages of adult mice per colony were sampled at random each quarter. Fecal samples from other supplier-origin colonies were collected from age- and sex-matched groups of mice.

Working in a biosafety cabinet, mice were transferred individually to an empty polycarbonate cage with no bedding and allowed to defecate naturally. Mice were then returned to their home cage, and fecal pellets were collected using autoclaved wooden toothpicks, which were discarded after each use. Fecal samples were placed in 2 mL round-bottom tubes containing a 0.5 cm diameter stainless steel bead. Tubes were transported to the lab and kept frozen until DNA extraction.

## DNA extraction

DNA was extracted using QIAamp PowerFecal Pro DNA extraction kits (Qiagen) according to the manufacturer's instructions, with the exception that samples were homogenized in bead tubes using a TissueLyser II (Qiagen, Venlo, Netherlands) for 10 min at 30 Hz, rather than using the vortex adapter described in the protocol. Samples were then processed according to the protocol and eluted in 100 µL of elution buffer (Qiagen). In January of 2021, Qiagen announced that the chemistry used in all QIAamp PowerFecal Pro DNA extraction kits moving forward would change immediately to accommodate incubation at room temperature and shorter incubation periods. All DNA yields were quantified via fluorometry (Qubit 2.0, Invitrogen, Carlsbad, CA) using quant-iT BR dsDNA reagent kits (Invitrogen) and normalized to a uniform concentration and volume.

## 16S rRNA library preparation and sequencing

Library preparation and sequencing were performed at the MU Genomics Technology Core. Amplicon libraries were constructed via amplification of the V4 region of the 16S rRNA gene with universal primers (U515F/806R), flanked by Illumina standard adapter

sequences (50, 51). Dual-indexed forward and reverse primers were used in all reactions. PCR was performed in 50 µL reactions containing 100 ng metagenomic DNA, primers (0.2 µM each), dNTPs (200 µM each), and Phusion high-fidelity DNA polymerase (1U, Thermo Fisher). Amplicon pools (5 µL/reaction) were combined, thoroughly mixed, and then purified by addition of Axygen Axyprep MagPCR clean-up beads to an equal volume of 50 µL of amplicons and incubated for 15 min at room temperature. Products were then washed multiple times with 80% ethanol, and the dried pellet was resuspended in 32.5 µL EB buffer (Qiagen), incubated for 2 min at room temperature, and then placed on the magnetic stand for 5 min. The final amplicon pool was evaluated using the Advanced Analytical Fragment Analyzer automated electrophoresis system, quantified using quant-iT HS dsDNA reagent kits, and diluted according to Illumina standard protocol for sequencing as $2 \times 250$ bp paired-end reads on the MiSeq instrument.

## Informatics procedures

Cutadapt (52) (version 2.6) was used to remove the primer from the 5′ end of the forward read. If found, the reverse complement of the primer to the reverse read was then removed from the forward read as were all bases downstream. Thus, a forward read could be trimmed at both ends if the insert was shorter than the amplicon length. The same approach was used on the reverse read, but with the primers in the opposite roles. Read pairs were rejected if one read or the other did not match a 5′ primer, and an error rate of 0.1 was allowed. Two passes were made over each read to ensure the removal of the second primer. A minimal overlap of 3 bp with the 3′ end of the primer sequence was required for removal. The QIIME2 (53) and DADA2 (54) plugin (version 1.10.0) was used to denoise, de-replicate, and count ASVs (amplicon sequence variants), incorporating the following parameters: (1) forward and reverse reads were truncated to 150 bases; (2) forward and reverse reads with number of expected errors higher than 2.0 were discarded; and (3) chimeras were detected using the "consensus" method and removed. R version 3.5.1 and Biom version 2.1.7 were used in QIIME2. Taxonomies were assigned to final sequences using the Silva.v132 (55) database with the classify-sklearn procedure.

## Statistical and machine learning analyses

For univariate outcomes (e.g., observed richness), data were first tested for normality and equal variance using the Shapiro-Wilk and Brown-Forsythe methods, respectively, and the appropriate parametric or nonparametric tests were applied. These included analysis of variance (ANOVA) or Kruskal-Wallis ANOVA on ranks, respectively, for comparisons of all four GMs, or Student's *t*-test or Fisher's exact test, respectively, for comparisons of two groups. Univariate statistical testing was performed using SigmaPlot 15.0 (Grafiti, LLC, Palo Alto, CA). Permutational multivariate ANOVA (PERMANOVA) was used to test for differences in unweighted and weighted beta-diversity. Differential abundance testing was performed using linear discriminant analysis (LDA) effect size analysis (LEfSe v1.1.01) (56). SourceTracker (version 2.0.1) (57) was used to predict the percent contribution of each supplier-origin gut microbiome (source) to the microbiome of GEM models (sink). Multivariate analyses were performed using Past 5.0 software or in R.

## RESULTS

### Microbiome of GEM lines is shaped primarily by proximal environmental factors

In total, our analysis incorporated data from 351 samples from genetically engineered mice (GEM) submitted to the MU MMRRC for cryopreservation. When possible, multiple fecal samples were collected and analyzed from mice of the same line. As such, these 351 samples represent 275 distinct GEM lines from 139 different investigators at 84 different institutions in 33 states, as well as Washington, D.C., Canada, France, Germany, Belgium, Australia, Japan, and Singapore. Across all samples, an average (±SE) of 41,565

(± 1,058) high-quality reads per sample was detected. To accommodate for variability in sequencing depth, data were rarefied to a uniform number (16,749) of sequence reads per sample, and singleton sequences were discarded.

The richness observed in fecal samples from mice submitted to the MMRRC for cryopreservation ($n$ = 351) demonstrated a wide range (52 to 520 ASVs, mean 268.7) with a sizeable tail (19/351 samples, 5.4%) on the low end of the distribution comprising less than 100 ASVs per sample (Fig. S1). To evaluate the relative contribution of environment at the level of lab and institution, the mean unweighted similarities between institutions, between labs within institution, between genetically distinct mouse lines within lab, and within mouse lines were compared (Fig. 1). On average, mice of the same strain as well as mice of different strains submitted by the same lab were more similar to each other than to mice from different labs at the same institution or from different institutions. Kruskal-Wallis ANOVA on ranks failed to detect a difference between inter-institution and intra-institution similarities, or between inter-strain and intra-strain similarities of mouse lines within the same lab, collectively indicating that the immediate laboratory environment is the dominant driver of beta-diversity in these data. Visualization of beta-diversity among GEM samples based on genetic background showed no reliable clustering (Fig. S2A). In instances of apparent clustering of samples from mice of the same genetic background (e.g., NOD samples), genetic background was confounded by the submitting lab. Comparison of GEMs submitted by 13 different labs at the University of Missouri (MU) and samples from external institutions demonstrated high intra-lab but low intra-institution similarity (Fig. S2B), reflecting our initial analysis.

## Microbiome of most GEM lines is distinct from supplier-origin microbiomes

Many of these GEM lines were generated using mice obtained from one of the four primary suppliers of mice in the U.S.A. As the mice from these suppliers harbor supplier-specific microbiomes, we hypothesized that the microbiome of GEMs would retain features indicative of their ancestral source(s). To test this, we assembled a reference data set consisting of 1,171 samples from CD-1, B6J, FVB, and $Apc^{min}$ mice colonized with one of four supplier-origin (SO) gut microbiomes (GMs). As previously observed, SO GMs differed in richness, with GM1, GM2, GM3, and GM4 all being significantly different from each other ($P$ < 0.001) with the exception of GM2 versus GM3 (Fig. 2A). Two-way ANOVA indicated a significant effect of background strain on richness. Specifically, GM4 demonstrated uncharacteristically low richness (mean 241 ASVs) when present in FVB mice (Fig. S3A). When present in B6J, CD1, or $Apc^{min}$ mice, GM4 had a mean richness of 325, 338, and 345 ASVs, respectively (Fig. S3B through D). Ordination of SO GM samples using unweighted dissimilarities revealed clear separation of SO GMs, with the exception of GM2 and GM3, which overlapped along principal coordinate (PC) 1 (Fig. 2B). Permutational multivariate ANOVA (PERMANOVA) indicated a significant difference among SO GMs ($P$ = 0.0001, F = 185.2). Pairwise comparisons confirmed significant differences between all SO GMs ($P$ = 0.0001) with F values ranging from 24 (GM2 vs GM3) to 438 (GM1 vs GM4). Two-way PERMANOVA including GM and background strain as factors detected significant effects of each, although the effect of GM ($P$ = 0.0001, F = 90) was greater than that of strain ($P$ = 0.0001, F = 25), as reflected by partial separation of samples by strain within each GM (Fig. S4). Recognizing that many ASVs are annotated to similar or closely related taxonomies, samples were ordinated using features collapsed to the level of genus (Fig. S5). While the distance between clusters decreased, groups continued to separate. PERMANOVA based on Jaccard dissimilarities using genus-level features yielded $P$ = 0.0001, F = 176, with similar outcomes in pairwise comparisons. To identify genus-level biomarkers of each SO GM, linear discriminant analysis (LDA) effect size (LEfSe) analysis was performed. Several genera were identified as biomarkers of each SO GM, with each supplier represented by members of the *Bacteroidota* and *Bacillota* as well as other well-recognized disease-modifying organisms, such as *Akkermansia*, *Mucispirillum*, and segmented filamentous bacteria (SFB, *Candidatus Arthromitus*) (Fig. S6).

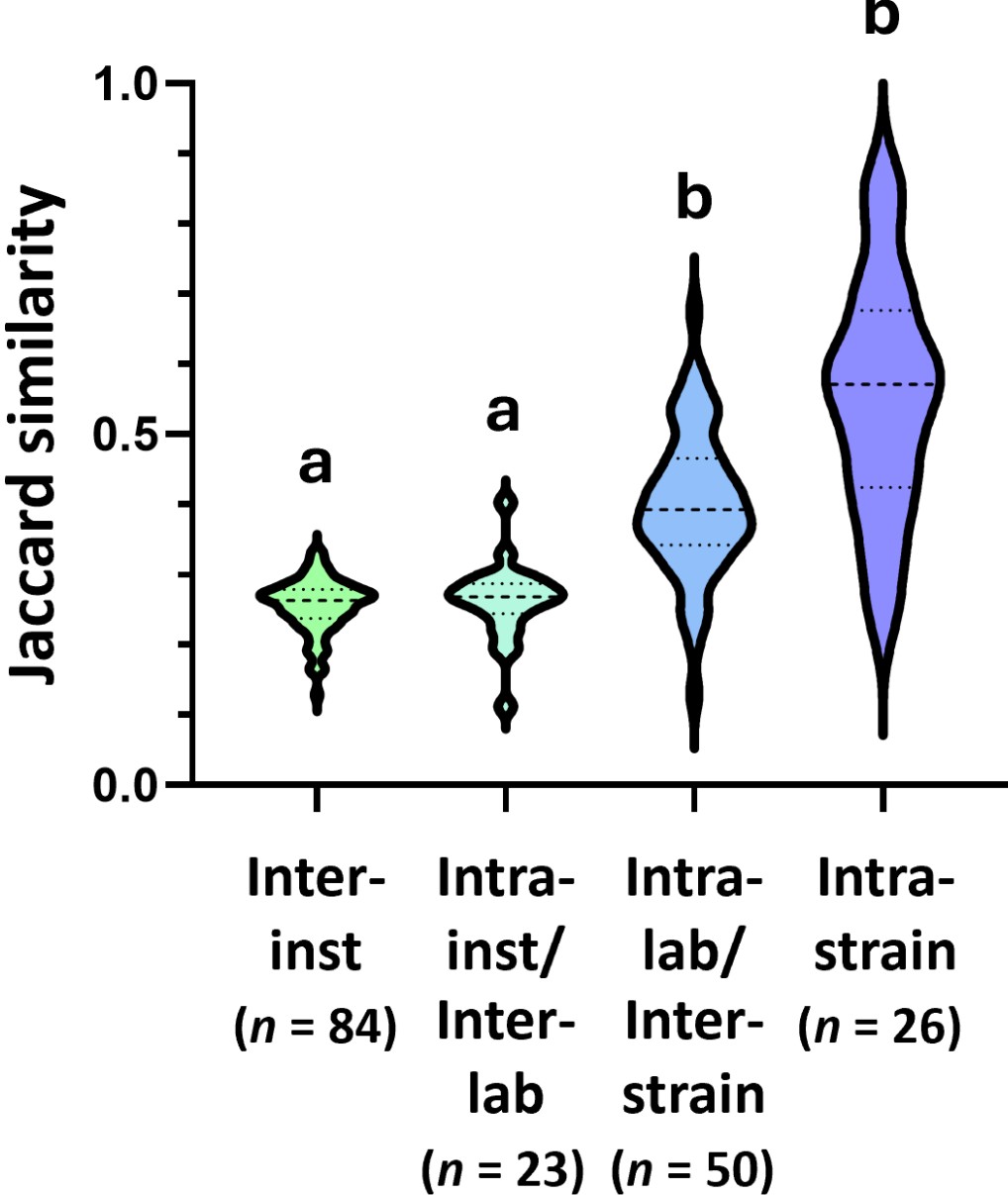

**FIG 1** Violin plot showing distribution of the mean similarity between institutions (inter-inst, *n* = 84), the mean similarity between different labs within the same institution (intra-inst/inter-lab, *n* = 23), the mean similarity between strains within the same laboratory (intra-lab/inter-strain, *n* = 50), and the mean within-strain similarity for cases wherein samples from multiple mice of the same strain (mean 3.57 per strain) were obtained (intra-strain, *n* = 26). Different letters indicate significant differences between groups based on one-way ANOVA. For Jaccard similarity, 1 = identical membership, 0 = mutually exclusive membership.

Concurrent ordination of data from SO GMs and cryopreserved GEMs revealed an unexpected clustering of GEM samples, primarily overlapping GM3 along PC1 and PC2 (Fig. 3A) but distinct from all four SO GMs when PC3 is included (Fig. 3B). A separate small group of GEM samples overlapped with GM1. PERMANOVA detected significant differences between GEMs and SO GMs (*P* = 0.0001, F = 141.5), including all pairwise comparisons. To reduce the contribution of SO GMs to beta-diversity while retaining their use as reference points, we repeated these analyses using a smaller, uniform number of samples (*n* = 20) from CD-1 mice with each SO GM. Samples from GEMs now fell into a distinct cluster partially overlapping GM4 and GM3, a small cluster overlapping

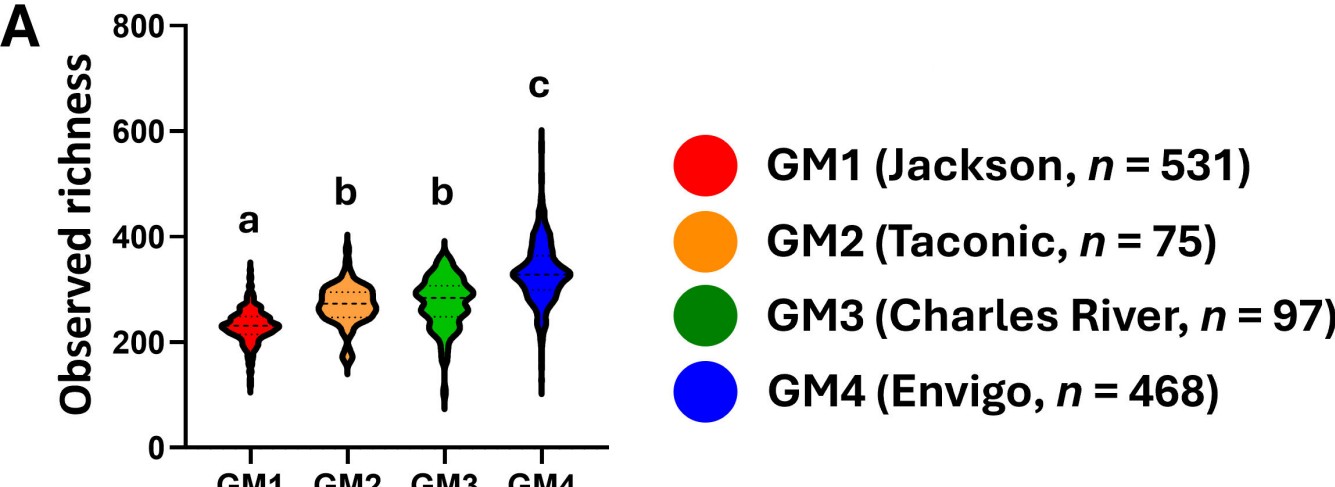

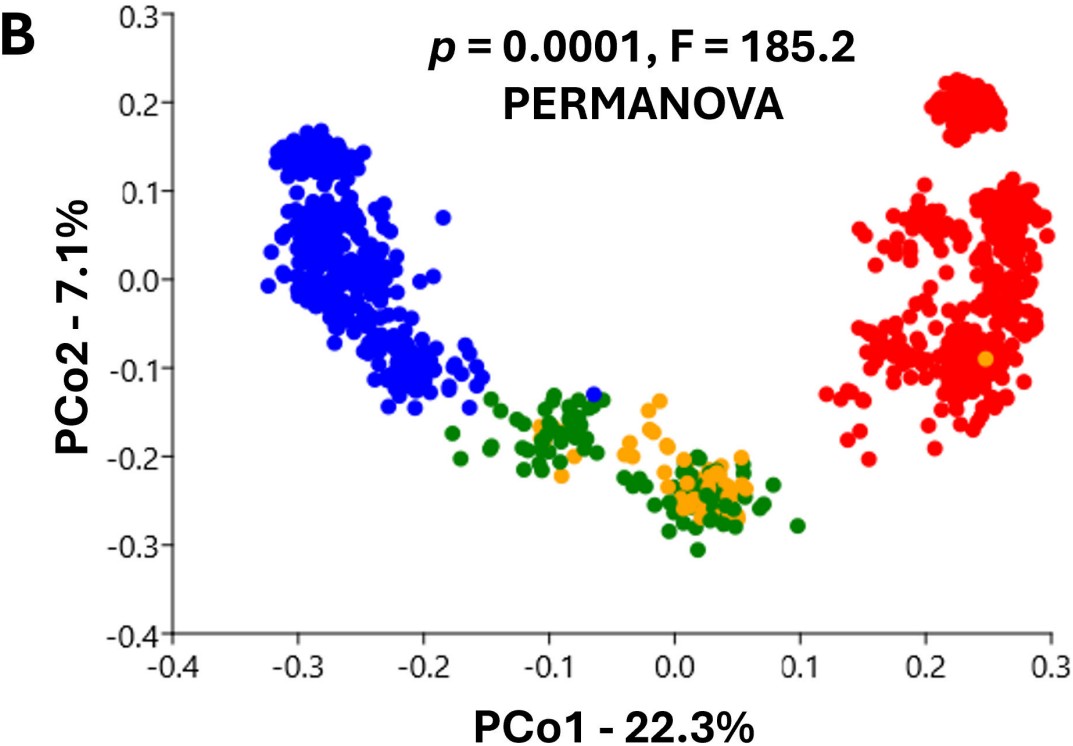

**FIG 2** (A) Violin plots showing observed richness in fecal samples from CD-1, B6J, Apc$^{min}$, and FVB mice colonized with one of four supplier-origin (SO) gut microbiomes (GM1 to GM4, origin of each and sample number in legend at right). Different letters indicate significant differences between SO GMs in Tukey *post hoc* comparisons following ANOVA, all $P < 0.001$. (B) Principal coordinate analysis (PCoA) plots showing unweighted beta-diversity among samples shown in panel **A**. Results of permutational multivariate ANOVA (PERMANOVA) based on unweighted dissimilarities are shown.

GM1, and a small cluster distinct from other GEMs and SO GMs (Fig. 3C). PC3 captured additional beta-diversity among GEMs (Fig. 3D). The reduced number of SO GM samples did not affect the conclusions of PERMANOVA testing ($P = 0.0001$, F = 13.7). Pairwise comparisons detected significant differences between GEMs and all SO GMs. While many GEM samples overlapped with GM1, GM3, and GM4 along PC1, the distribution of samples from GEMs suggested a large contribution to beta-diversity of features independent of SO GMs.

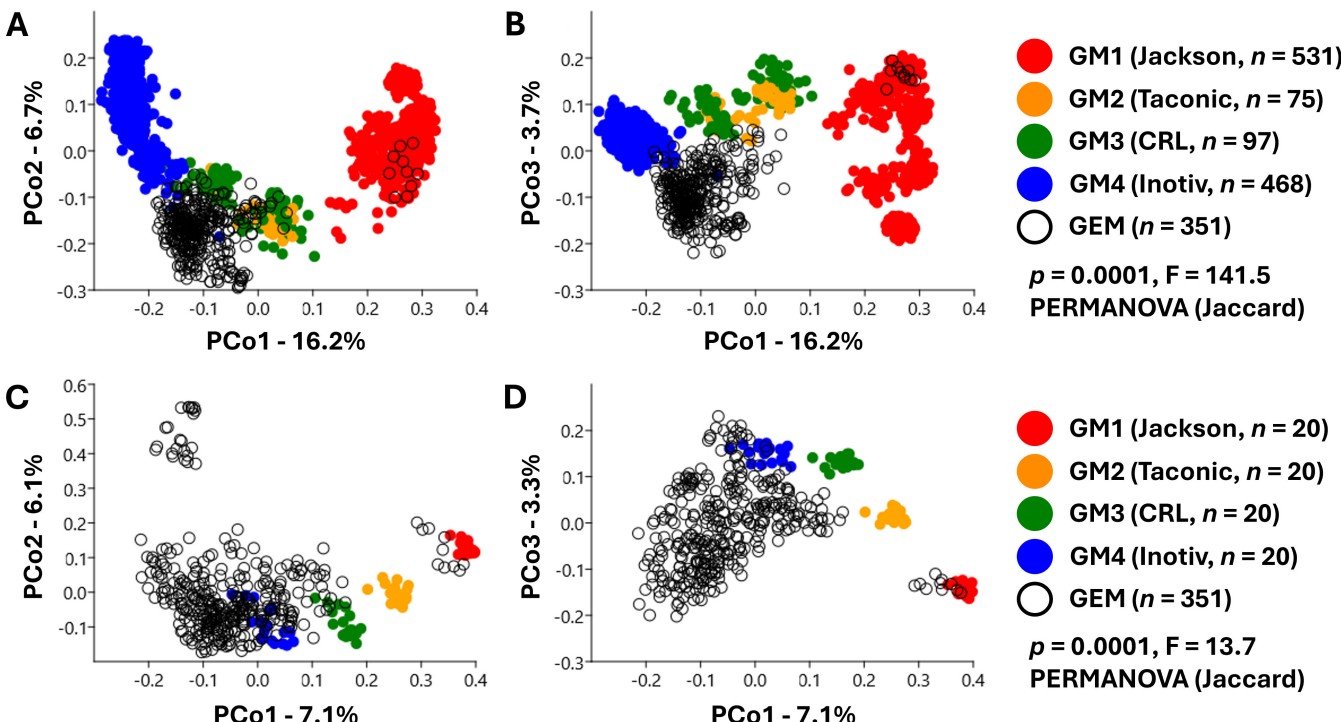

**FIG 3** (A, B) Principal coordinate analysis (PCoA) plots showing unweighted beta-diversity along PCo1 and PCo2 (A) and PCo3 (B) among samples from supplier-origin (SO) gut microbiome (GM)-colonized mice of multiple genetic backgrounds (*n* = 1,171, legend at right) and genetically engineered mouse (GEM) models submitted to the MU MMRRC (*n* = 351). (C, D) PCoA plots showing unweighted beta-diversity along PCo1 and PCo2 (C) and PCo3 (D) among 80 samples from SO GM-colonized CD-1 mice (*n* = 20 per supplier, legend at right) and GEM models (*n* = 351). P and F values from one-way permutational multivariate analysis of variance (PERMANOVA) using Jaccard dissimilarities.

## Much of the microbiome of GEM lines is derived from unknown sources

To more fully explore the contribution of SO GMs to the microbiome of GEMs and assess their utility in classification of the GM of GEMs (or any unknown GM), we performed a SourceTracker analysis. SourceTracker (57) applies Bayesian statistical methods to estimate the relative contribution of different possible "source" microbial communities (i.e., SO GMs) within "sink" communities of unknown origin (i.e., GEMs), as well as the relative contribution of unknown features not detected in any of the possible sources. In the majority of GEM samples, unknown sources made the largest contribution to community composition, followed by GM2 and GM3 (Fig. 4A; Fig. S7). A substantial contribution by multiple SO GMs was detected in most GEM samples. Figure 4B shows a cladogram, resolved to the level of genus, of all taxa detected in GEMs. The outer rings of the cladogram indicate the presence or absence of the genus in each of the SO GMs. While most families and genera detected in GEMs are represented in at least one of the SO GMs, there are several genera unique to GEMs and not found in any of the 1,171 SO GM samples. Full taxonomies and their mean (±SD) relative abundance (RA) in GEMs and each SO GM are provided in Table S1.

## Enterohepatic *Helicobacter* spp. are common in GEM lines

At face value, these findings presented a paradox. The data from GEMs submitted by 139 investigators at 84 different research institutions suggest that immediate (laboratory-level) factors have a dominant influence on the composition of the fecal microbiota. At the same time, however, the microbiome of most GEMs demonstrated a surprisingly high similarity to each other. Data were collapsed to the level of genus and manually curated to identify features that were common among GEMs but rare or not detected in SO GMs.

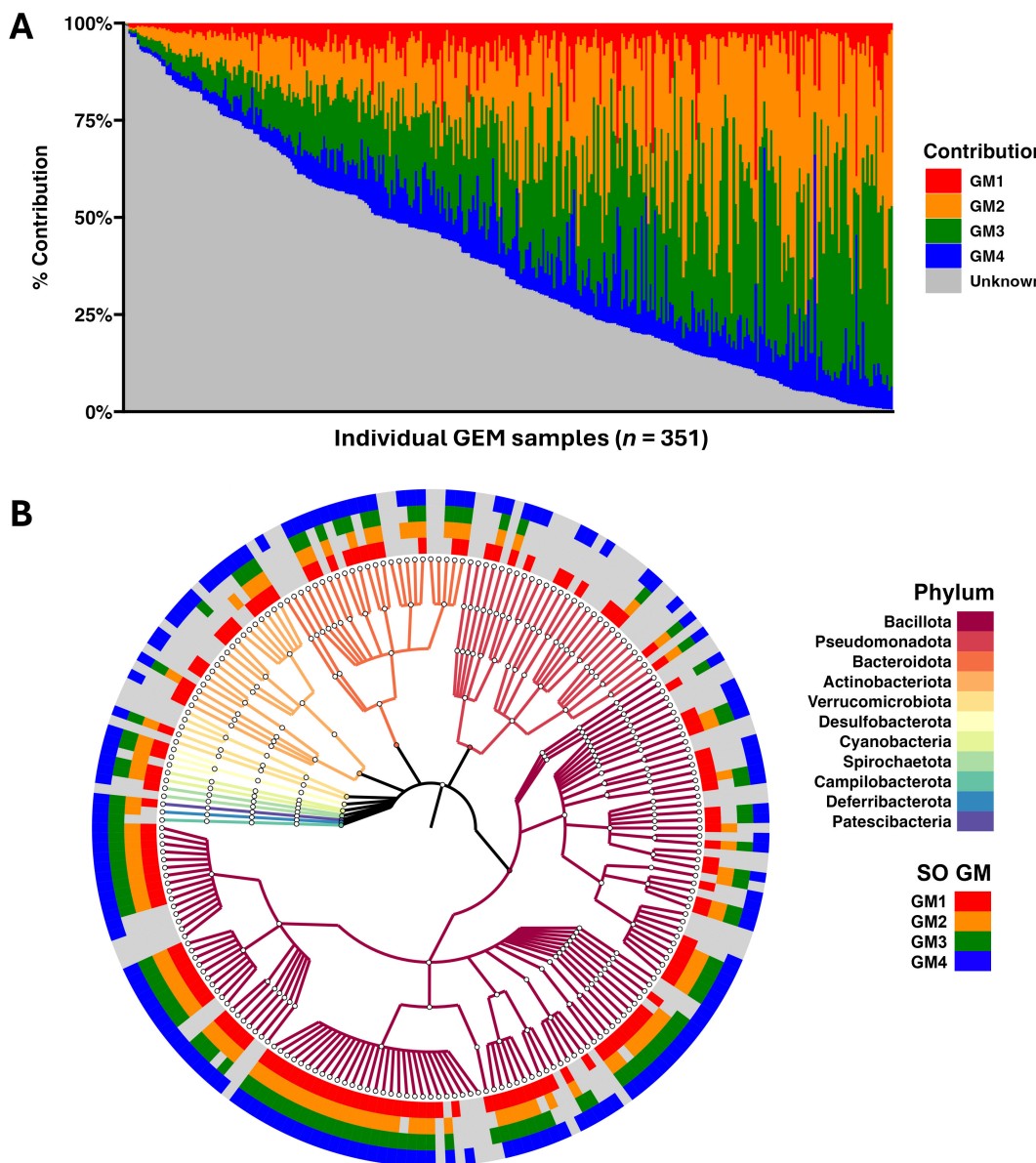

**FIG 4** (A) Stacked bar chart showing the predicted percent contribution of each supplier-origin (SO) gut microbiome (GM, legend at right) to the microbiome of genetically engineered mouse (GEM) models (*n* = 351) ranked in decreasing order of percent contribution from unknown sources, as determined using SourceTracker2. (B) Cladogram colored according to phylum (legend upper right) showing genera (outer spokes) identified in GEMs. Outer rings are colored according to the presence or absence of each genera among the supplier-origin gut microbiome training data (legend lower right).

Of the 71 genus-level features identified in >50% of the GEM samples, the only feature rarely detected in any of the SO GMs was *Helicobacter* sp.

In total, sequences annotated to the genus *Helicobacter* were detected in 7 of 1,171 (0.59% prevalence) SO GM samples (as single or double reads, i.e., <0.012% relative abundance), and none of those sequences was annotated to the level of species. In contrast, *Heliobacter* sp. was detected in 74.6% of GEM samples, at a mean relative abundance of 1.78%. Moreover, the majority of these sequences were resolved to the level of species, including (in decreasing order of prevalence) *H. hepaticus*, *H. typhlonius*, *H. mastomyrinus*, *H. apodemus*, *H. bilis*, and *H. rodentium* (Fig. S8A). Of the 351 GEM samples, a single species was detected in 127 samples, while multiple *Helicobacter* spp. were detected in 126 samples, with as many as five different species co-infecting some mice (Fig. S8B and C).

## Relative abundance (RA) of *Helicobacter* spp. correlates with RA of other mucosal colonizers

As *Helicobacter* spp. are capable of colonizing the inner mucus layer, correlation analyses were performed between genus-level abundance of *Helicobacter* spp. and several other SO GM biomarker bacteria capable of colonizing the inner mucus layer, including *Mucispirillum* (phylum *Deferribacterota*), segmented filamentous bacteria (SFB, phylum *Bacillota*), *Desulfovibrio* and *Bilophila* (phylum *Thermodesulfobacteriota*), and *Akkermansia* (phylum *Verrucomicrobiota*) spp. (Fig. 5A). Significant positive correlations were detected between the colonization level of *Helicobacter* and *Mucispirillum* ($R^2 = 0.404$, $P = 6.66 \times 10^{-16}$), SFB ($R^2 = 0.263$, $P = 6.58 \times 10^{-7}$), *Desulfovibrio* ($R^2 = 0.296$, $P = 1.88 \times 10^{-8}$), and *Bilophila* ($R^2 = 0.350$, $P = 2.01 \times 10^{-11}$). In contrast, significant negative correlations were detected between *Akkermansia* (a GM1 biomarker) and *Helicobacter* ($R^2 = -0.214$, $P = 5.45 \times 10^{-5}$), *Mucispirillum* ($R^2 = -0.146$, $P = 0.0062$), SFB ($R^2 = -0.170$, $P = 0.00137$), and *Desulfovibrio* ($R^2 = -0.157$, $P = 0.00314$). Members of the *Bacteroidota* are particularly dependent on host mucins as a source of amino acids (58). As one or more genera within the *Bacteroidota* were identified among the top biomarkers for each SO GM, correlation analyses were also performed between *Helicobacter* and these biomarker genera (*Bacteroides*, GM1; *Muribaculum*, GM2; UC *Muribaculaceae*, GM3; and *Alloprevotella* and *Rikenella*, GM4). Significant positive correlations were detected between *Helicobacter* and both *Alloprevotella* ($R^2 = 0.442$, $P = 2 \times 10^{-7}$) and *Rikenella* ($R^2 = 0.407$, $P = 2 \times 10^{-7}$), but none of the other *Bacteroidota* biomarkers tested (Fig. 5B). These correlations suggest an association between *Helicobacter* colonization and features of GM4 (SFB, *Desulfovibrio*, *Bilophila*, *Alloprevotella*, and *Rikenella*) and certain other taxa.

## DISCUSSION

For the sake of reproducibility, it is necessary to include details related to the genotype of mouse models used in research, including not just genetic modifications, but also the full host genetic background. An undeniable body of evidence, however, reveals the profound influence of the GM on host development and homeostasis with relevance to virtually every medical specialty. Thus, the use of animal models involves holobiont organisms, the full genotype of which is reflected in the collective host genome and metagenome.

With this in mind, the data presented here are noteworthy for several reasons. First, these data show that only a small portion of the variability among the microbiome of genetically engineered mouse (GEM) models is explained by features originating from the four domestic suppliers of laboratory mice in the U.S.A. While we recognize that the SO GMs used for reference data do not capture the entirety of SO GMs in mice from other rooms or production facilities, the high prevalence (75%) of microbes, such as *Helicobacter* spp., that are excluded from most SPF mice suggests the introduction of taxa from other sources. This is further supported by results indicating a dominant effect of microenvironment (laboratory) compared to macroenvironment (institution) or genetic background. Thus, while the supplier is a strong determinant of the laboratory mouse GM, immediate environmental factors have a comparable, if not greater, effect on the lab mouse GM at the colony level.

Second, the substantial contribution of microbes of unknown origin has direct and indirect implications on model performance and reporting. Differences between SO GMs are associated with phenotypic effects in a broad range of mouse disease models (24, 26–28, 31, 59) and normative development and homeostasis (36, 37, 60, 61). Similarly, facility-dependent differences in the GM within the same institution are sufficient to induce significant changes in the mucus layer of the hindgut (62), further affecting accessibility of mucin compounds to taxa in the GM. Our current data suggest that it is insufficient to rely on the commercial source of mice reported in scientific literature to deduce the composition of their GM. For colonies established and maintained at research institutions, it is likely that the GM of mice has acquired additional members. This puts the onus of colony microbiome surveillance and reporting on individual investigators. As

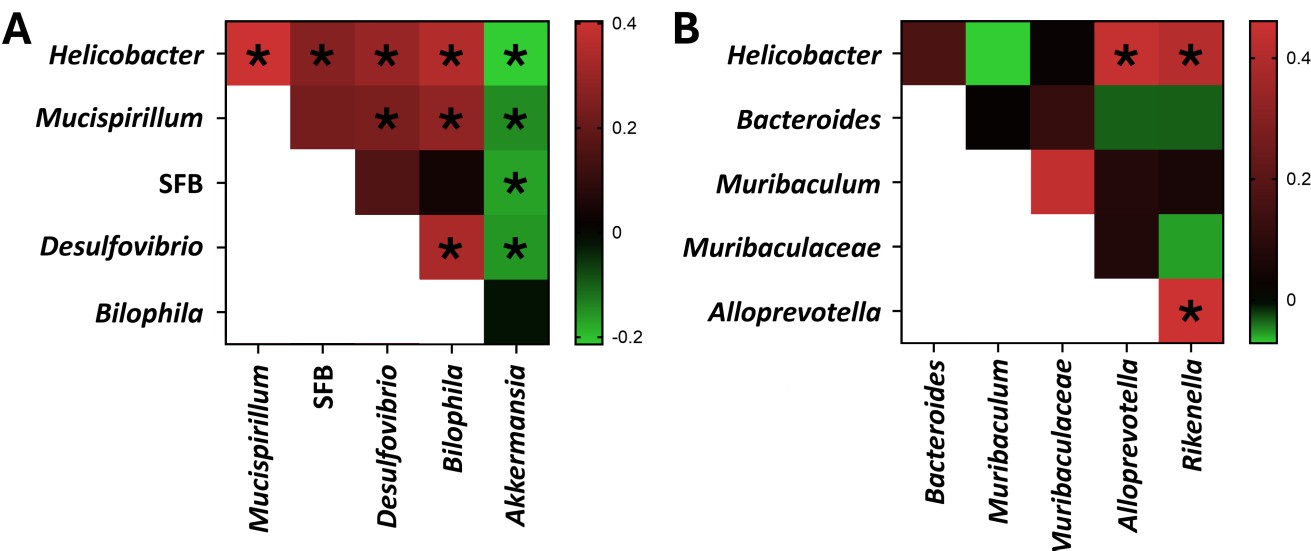

**FIG 5** Heatmaps showing the correlation ($R^2$, Spearman correlation, legend at right) in relative abundance of select biomarker genera from phyla other than *Bacteroidota* (A) and biomarker taxa within *Bacteroidota* (B) among samples from GEMs (*n* = 351 mice, 275 distinct lines) submitted from 139 different investigators to the MMRRC for cryopreservation. Asterisks denote significance ($P < 0.05$), Spearman's rank correlation.

this carries additional costs, surveillance and reporting of the microbiome of research mice should not necessarily be made requisite by journals or funding agencies, but such efforts should be rewarded by reviewers as a demonstration of scientific rigor. These data also highlight the challenges associated with long-term maintenance and monitoring of colony GMs due to unavoidable factors like availability of feed and bedding.

*Helicobacter* spp. are members of the native GM of wild mice, associated with protective effects in disease models, wherein CD8+ cytotoxic lymphocytes (CTL) would confer protection, such as viral infection and neoplasia (63). Murine *Helicobacter* spp. also induce antibody responses against resident microbes (64, 65), confer colonization resistance against enteric pathogens (48), and elicit a $T_H1$ immune response from CD4+T helper cells following experimental inoculation of naïve mice (66, 67). In other disease models, however, the robust CTL immunity induced by the wild mouse GM is detrimental to host fitness (68). Similarly, the acute immune response to *Helicobacter* spp. is used as the trigger for intestinal disease in several different genetically susceptible mouse models of inflammatory bowel disease (41, 69). Thus, enterohepatic *Helicobacter* spp. are disease-modifying organisms (DMOs) capable of exacerbating or mitigating disease severity depending on the pathogenic mechanism. In a comprehensive pathogen prevalence report from Charles River Laboratories spanning 2003 to 2020 (49), 13.8% of submitted pools tested positive for *Helicobacter* sp. using generic primers. The current data suggest that the prevalence of enterohepatic *Helicobacter* spp. in biomedical research facilities may be higher than detected in sentinel testing. Surveys of SPF and conventionally housed research mice performed in Asia, Europe, and North America roughly 20 years ago reported prevalence of enterohepatic *Helicobacter* spp. between 58% and 88% (70–73). More recent work mining publicly available metagenomic data from 1,091 SPF mice and 2,036 conventionally housed mice (from a total of 50 studies) reported a prevalence of 2.3% and 24.9%, respectively (48).

Colonization with enterohepatic *Helicobacter* spp. induces changes in community composition. Inoculation of Altered Schaedler Flora (ASF)-colonized mice with *H. bilis* favors or limits the abundance of certain members of ASF in apparently or competitive relationships while having limited effect on other members (74). Similar effects were reported following inoculation of C57BL/6J mice inoculated with *H. hepaticus* (75). The significant positive and negative correlations between *Helicobacter*-specific read

counts and counts for several other microbes potentially sharing the same mucosal environment may reflect similar mechanisms at work. While speculative, the presence of *Helicobacter* spp. may also partially explain the limited beta-diversity among most GEMs.

The greater similarity between strains submitted by the same laboratory, compared to strains from the same institution, suggests the effect of select environmental factors that may vary between rooms or facilities within an institution. As such, the immediate laboratory environment likely represents the cumulative effect of diet, water treatment, bedding, caging system, and a multitude of other husbandry-related factors. Importantly, the proximal environment is also influenced by biosecurity protocols and compliance among personnel. While it is unclear how *Helicobacter* spp. were introduced into the microbiome of the GEM lines in the current study, we speculate that it was more likely obtained through direct contact with pre-existing mice being used in the laboratory (e.g., breeding to cross lines) rather than through biosecurity breaches resulting in exposure to sources such as wild mice or other mice in the same room. Importantly, enterohepatic *Helicobacter* spp. are non-spore-forming microaerobic organisms that are transmitted via the fecal-oral route and do not survive long outside of the host. Basic containment, such as filter-top caging, prevents its spread within a room (72), and *Helicobacter* spp. are susceptible to common disinfectants (76, 77). Additionally, while the prevalence of *Helicobacter* spp. approaches 100% in wild mice (48, 78, 79), the aforementioned literature suggests that *Helicobacter* spp. are also present in many conventionally housed research mice (48, 71, 73). Several scenarios could explain the colonization of GEM lines. For example, even transient contact (e.g., timed matings) between newly created GEM lines and *Helicobacter*-colonized mice from other lines allows transmission via coprophagy. Alternatively, GEMs generated in facilities using *Helicobacter*-infected mice (e.g., as surrogate dams for embryo transfer procedures) could also explain the high prevalence.

While collective analysis of the GEM data clearly shows the strong effect of immediate environmental factors on the microbiome, certain genotypes may also have an influence. Examples include the distinct cluster of samples from GEMs on a CBA genetic background (Fig. S2A). Of note, however, the three GEM samples from mice on an SJL background clustering near the CBA mice are from a different lab within the same institution, again suggesting the possibility of an environmental factor. Interestingly, the four GEM samples on a B6J.129SvEv genetic background clustering near the aforementioned CBA and SJL mice are Bbs4 mutants created using gene trap technology (80). Several other mice on similar genetic backgrounds, including concurrent Bbs1 mutants on an identical background from the same lab, are included within the larger cluster of GEM samples. Bbs1 and Bbs4 both encode proteins that are part of the BBSome protein complex involved in primary cilia homeostasis and function. Considering the broad range of neurodevelopmental abnormalities observed in primary ciliopathies (81), effects on the gut microbiome would not be surprising. Similarly, the GEMs from MU lab C that clustered distinctly from other GEMs from the same lab differed primarily in their genetic background. IL13ra1-deficient and IL-13ra1/EGFP double-knockout mice on BALB/cJ ($n = 5$), CJ.129S2 ($n = 3$), and SJL ($n = 2$) backgrounds are all contained in the main cluster of GEM samples, whereas mice on a B6 background ($n = 4$) cluster distinctly from the main group of GEMs and other strains from the same lab, suggesting an interaction between the genetic modification and background. Although the current data are not properly powered to interrogate the effect of specific genotypes, they may nonetheless provide insights regarding the gut microbiome of specific GEMs, and raw data are publicly available as a resource to investigators.

The institutions submitting samples are representative of a broad swath of the biomedical research landscape, comprising multiple branches of over 50 public and private academic institutions (including 32 AAU members and 13 land-grant universities), nine non-profit medical research institutions, eight international research institutions, four NIH Institutes, and three comprehensive cancer centers. The fact that the immediate laboratory environment has such a profound effect on the GM of these

samples suggests that survey and reporting of the GM is most appropriately performed by investigators at the time of phenotyping. While a small number of GEM samples in the current study mirrored the Jackson-origin GM1, the majority of samples could not be reliably classified according to any of the SO GMs, indicating the need for more granular reporting. The current era of scientific discovery offers investigators numerous resources to survey or characterize the GM of animal models used in their research. Metabarcoding and other platforms are offered commercially or through core facilities at many institutions. At a minimum, it seems reasonable to consider diagnostic testing via PCR of not just sentinels but also experimental animals for *Helicobacter*, SFB, or other DMOs for which commercially available testing becomes available.

Perhaps counterintuitively, we speculate that colonization with enterohepatic *Helicobacter* spp. may actually enhance the translatability of data generated using GEMs, not due to a more faithful recapitulation of the human microbiome, but rather due to inclusion of a dominant member of the native microbiome of wild mice required for normal host physiology. Enterohepatic *Helicobacter* spp. are almost ubiquitous in the wild mouse microbiome (79) but are uncommon in human fecal samples and often associated with adverse conditions when detected (82, 83). Compelling studies using lab mice rederived to harbor the microbiome of wild mice have demonstrated immune characteristics more similar to those of adult humans (84) and improved better predictive power during drug development (68). While these effects cannot be attributed solely to *Helicobacter* spp., it is clearly a dominant member of the wild mouse microbiome with recognized effects on innate and adaptive immunity (65, 85, 86) and colonization resistance (48).

Limitations of the current study include the inability to capture the entirety of SO GMs available through domestic suppliers. Indeed, much of the unknown portion of GEM microbiomes may represent taxa found in mice at other facilities than those from which mice were obtained here. As mentioned before, however, the high prevalence of *Helicobacter* spp. among GEMs suggests external sources contribute to the beta-diversity of these samples. It is difficult to completely separate the effects of background genotype from laboratory due to the overlap in some features. That being said, the current data contains sufficient different institutions, labs within institutions, and strains within labs to provide a meaningful comparison of mean similarity at each level.

In summary, the data presented here show that the GM of mice used in biomedical research commonly contains a substantial contribution from unknown sources, including *Helicobacter* spp. The presence of *Helicobacter* spp. is associated with the differential abundance of several other genera recognized to influence model phenotypes. These data also highlight the challenges associated with maintaining vivarium biosecurity and the value of surveying the GM of experimental mouse colonies at the time of phenotyping.

## AUTHOR AFFILIATIONS

[1]Pathobiology and Integrative Biomedical Sciences, University of Missouri, Columbia, Missouri, USA

[2]Mutant Mouse Resource and Research Center at the University of Missouri (MU MMRRC), Columbia, Missouri, USA

[3]University of Missouri Metagenomics Center (MUMC), Columbia, Missouri, USA

[4]IDEXX BioAnalytics, Columbia, Missouri, USA

## AUTHOR ORCIDs

Aaron C. Ericsson http://orcid.org/0000-0002-3053-7269

## FUNDING

| Funder | Grant(s) | Author(s) |
|---|---|---|
| NIH Office of the Director | U42 OD010918 | Aaron C. Ericsson |
| | | Zachary L. McAdams |
| | | Rebecca A. Dorfmeyer |
| | | Armedia O'Neill-Blair |
| | | James Amos-Landgraf |
| | | Craig L. Franklin |

## AUTHOR CONTRIBUTIONS

Aaron C. Ericsson, Conceptualization, Data curation, Formal analysis, Funding acquisition, Investigation, Methodology, Supervision, Visualization, Writing – original draft, Writing – review and editing | Zachary L. McAdams, Data curation, Formal analysis, Methodology, Visualization, Writing – review and editing | Rebecca A. Dorfmeyer, Investigation, Methodology, Writing – review and editing | Marcia L. Hart, Investigation, Methodology, Writing – review and editing | Armedia O'Neill-Blair, Investigation, Methodology, Writing – review and editing | James Amos-Landgraf, Conceptualization, Formal analysis, Funding acquisition, Methodology, Resources, Writing – review and editing | Craig L. Franklin, Conceptualization, Funding acquisition, Investigation, Methodology, Resources, Supervision, Writing – review and editing

## DATA AVAILABILITY

All sequencing data and metadata supporting the analyses below are publicly available at the National Center for Biotechnology Information (NCBI) Sequence Read Archive (SRA) as BioProject PRJNA1013504 (GEM lines) and PRJNA1334457 (MMRRC colony surveys).

## ETHICS APPROVAL

All animal work was performed under the full approval of the University of Missouri (MU) Institutional Animal Care and Use Committee (IACUC), protocol 39969.

## ADDITIONAL FILES

The following material is available online.

### Supplemental Material

**Supplemental figures (mSystems01112-25-s0001.pdf).** Figures S1 through S8.
**Table S1 (mSystems01112-25-s0002.xlsx).** Mean relative abundance of taxa detected in GEM lines and supplier-origin microbiomes.

### Open Peer Review

**PEER REVIEW HISTORY (review-history.pdf).** An accounting of the reviewer comments and feedback.

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
