## [Reviewer comments · mSystems]

Dominant effects of the immediate environment on the gut microbiome of mice used in biomedical research

Aaron Ericsson, Zachary McAdams, Rebecca Dorfmeier, Marcia Hart, Armedia O'Neill-Blair, James Amos-Landgraf, and Craig Franklin

Corresponding Author(s): Aaron Ericsson, University of Missouri

Review Timeline:

Submission Date:	July 25, 2025
Editorial Decision:	September 5, 2025
Revision Received:	September 29, 2025
Editorial Decision:	October 10, 2025
Revision Received:	October 15, 2025
Accepted:	October 22, 2025

Editor: Jonathan Klassen

Reviewer(s): Disclosure of reviewer identity is with reference to reviewer comments included in decision letter(s). The following individuals involved in review of your submission have agreed to reveal their identity: Axel Kornerup Hansen (Reviewer #1); Jeffrey Price (Reviewer #2)

Transaction Report:

DOI: <https://doi.org/10.1128/msystems.01112-25>

Re: mSystems01112-25 (Dominant effects of the immediate environment on the gut microbiome of mice used in biomedical research)

Dear Dr. Aaron Conrad Ericsson:

As you can see, the reviewers had largely minor concerns that mostly have to do with clarifying the methods and results. I look forward to seeing a revised version of the manuscript with these points addressed.

Revision Guidelines

Sincerely,
Jonathan Klassen
Editor
mSystems

Reviewer #1 (Comments for the Author):

This manuscript by Ericsson et al is relevant, and the setup with four GM's and all these gene modified mice is unique. The result is not that surprising, but it always important to have documentation for our assumptions. AS underlined in the discussion reproducibility of mouse studies would need much more consideration of the GM.

I mostly have some comments to the form of the manuscript.

Ln 52-54 This sentence is a strong statement, which is correct, but one or several references are needed.

Line 107-115 Information on how the gene-modified mice got the GM is too sparse. Was FMT applied, was it by co-housing or???

Ln 122 Why were the GM's termed high and low?

Ln 127-134 The information on the mice and their housing is very sparse. Which brand was the cages, bedding etc. Which cage type were included in the IVC's, what was the stocking density, which environmental enrichment was given, what was temperature, humidity etc. I know we are talking about a long period, but there must be some standards in the unit.

Also you cannot test for 'all known pathogens'. Write, which ones, eventually in a supplementary file, or just write the name of the IDEXX profile.

Ln 193-194 Which tests were used for normality and equality? Which software was used for statistics?

'Results' It would make it easier to comprehend if subheading indicating the outcomes were used.

'Figure 1' The legend should be a bit more explanatory in relation to the meaning of intra- and inter. It is far easier to understand when reading the text in the manuscript.

'Figure 3' This could be two 3D-plots rather than four 2D-plots

Ln 289 *Helicobacter* in italics

'Figure 5A' Make the heat maps triangular to avoid double reporting. It makes it easier for the reader.

Ln 330-332 This sentence is a statement, which should have one or several references, although it is correct.

Reviewer #2 (Comments for the Author):

The manuscript by Ericsson, et al. is well-written and timely for microbiome research. Research is often impacted by the microbiome present in genetically-engineered mice, especially in experimental reproducibility between laboratories, but has not been often carefully scrutinized until recent years. Describing the major effectors of microbiome composition is important for analysis of all animal research. The authors' prescription that the GM of GEM is an important characteristic to be recorded at the time of phenotyping is well supported and a critical one for both reproducibility and translation to human research. The data analysis is appropriate and thorough. While this paper adds substantially to the microbiome literature, there are several points that should be addressed before publication.

Major points:

1. "Laboratory environment" needs to be defined. Although the manuscript mentions diet in the introduction, no data is presented on the background of bedding/food/etc. for GEM lines submitted to the MU MMRRRC. Although this metadata may not be available, intra-lab similarities could be largely driven by diet decisions for each laboratory, which would also implicate the differences in intra-institution GMs. This type of conclusion is important for their discussion of implications of this work. Biosecurity in facilities may be easier or more difficult based on the source of microbes, and understanding action elements prescribed by this paper depend on which of these inputs is responsible for the variance. Presumably all of this is meant to be included under "laboratory environment" but what the authors intend to include within that catchall needs to be explicitly stated or more thoroughly discussed.
 - a. Some of this may be able to be determined from local data- for instance MU lab C has split beta diversity based on whether a strain is B6/B6J or CJ_129 (Figure S2). Were those submissions split by time and or husbandry techniques as well, or can this be attributed mainly to mouse strain?
2. The characteristics of institutions submitting animals to MU MMRRRC should be interrogated to determine if this dataset is representative of GEM lines in research facilities in general (i.e. is there a confounding variable associated with these institutions?). This could either be through a short analysis of publicly available data for *Helicobacter* species within mouse lines, or through analysis of published data. Setting this data within the larger community, given the lack of *Helicobacter* presence in SO GM is important. What other research settings lack *Helicobacter* in their colonies?
3. Figure S8. The prevalence of *Helicobacter* found in these GEM lines is similar to what has been previously observed in laboratory mice in Europe, which correlated with prevalence in local wild mice (Wasimuddin, Applied and Environmental Micro, 2012). Because of the high importance the authors place on *Helicobacter* in GEM GM as compared to SO GM, possible sources

of Helicobacter should be discussed.

4. Figure S4. The interaction effect has a high negative value for F. This seems to indicate that the total variance is lower than that observed within the GM and/or strain values. The significant interaction effect should be better explained in the text in light of this value.

Minor points:

1. No IACUC statement.

2. Line 324- missing close parenthesis

3. Figure S3. No legend at right as stated in figure legend.

4. Figure S3. Extra letter c behind vertical label in D.

5. Although the authors mention reproducibility multiple times, the other major reason that GM is important in animal experiments is translatability. This manuscript would be strengthened by a comparison to human microbiomes, as helicobacter is implicated not just in models but in human disease. Could the authors do a similar correlation to Figure 5 of helicobacter with important bacteroidota in publicly available human microbiome data? Although not necessary, it would increase the impact of this work.

The manuscript by Ericsson, et al. is well-written and timely for microbiome research. Research is often impacted by the microbiome present in genetically-engineered mice, especially in experimental reproducibility between laboratories, but has not been often carefully scrutinized until recent years. Describing the major effectors of microbiome composition is important for analysis of all animal research. The authors' prescription that the GM of GEM is an important characteristic to be recorded at the time of phenotyping is well supported and a critical one for both reproducibility and translation to human research. The data analysis is appropriate and thorough. While this paper adds substantially to the microbiome literature, there are several points that should be addressed before publication.

Major points:

1. "Laboratory environment" needs to be defined. Although the manuscript mentions diet in the introduction, no data is presented on the background of bedding/food/etc. for GEM lines submitted to the MU MMRRRC. Although this metadata may not be available, intra-lab similarities could be largely driven by diet decisions for each laboratory, which would also implicate the differences in intra-institution GMs. This type of conclusion is important for their discussion of implications of this work. Biosecurity in facilities may be easier or more difficult based on the source of microbes, and understanding action elements prescribed by this paper depend on which of these inputs is responsible for the variance. Presumably all of this is meant to be included under "laboratory environment" but what the authors intend to include within that catchall needs to be explicitly stated or more thoroughly discussed.
 - a. Some of this may be able to be determined from local data- for instance MU lab C has split beta diversity based on whether a strain is B6/B6J or CJ_129 (Figure S2). Were those submissions split by time and or husbandry techniques as well, or can this be attributed mainly to mouse strain?
2. The characteristics of institutions submitting animals to MU MMRRRC should be interrogated to determine if this dataset is representative of GEM lines in research facilities in general (i.e. is there a confounding variable associated with these institutions?). This could either be through a short analysis of publicly available data for Helicobacter species within mouse lines, or through analysis of published data. Setting this data within the larger community, given the lack of Helicobacter presence in SO GM is important. What other research settings lack Helicobacter in their colonies?

3. Figure S8. The prevalence of Helicobacter found in these GEM lines is similar to what has been previously observed in laboratory mice in Europe, which correlated with prevalence in local wild mice (Wasimuddin, Applied and Environmental Micro, 2012). Because of the high importance the authors place on Helicobacter in GEM GM as compared to SO GM, possible sources of Helicobacter should be discussed.
4. Figure S4. The interaction effect has a high negative value for F. This seems to indicate that the total variance is lower than that observed within the GM and/or strain values. The significant interaction effect should be better explained in the text in light of this value.

Minor points:

1. No IACUC statement.
2. Line 324- missing close parenthesis
3. Figure S3. No legend at right as stated in figure legend.
4. Figure S3. Extra letter c behind vertical label in D.
5. Although the authors mention reproducibility multiple times, the other major reason that GM is important in animal experiments is translatability. This manuscript would be strengthened by a comparison to human microbiomes, as helicobacter is implicated not just in models but in human disease. Could the authors do a similar correlation to Figure 5 of helicobacter with important bacteroidota in publicly available human microbiome data? Although not necessary, it would increase the impact of this work.

The authors greatly appreciate the Reviewers' careful reading of our manuscript and their thoughtful suggestions. We address each of the Reviewer's comments individually below.

Reviewer #1:

Comment: This manuscript by Ericsson et al is relevant, and the setup with four GM's and all these gene modified mice is unique. The result is not that surprising, but it always important to have documentation for our assumptions. As underlined in the discussion reproducibility of mouse studies would need much more consideration of the GM.

Response: We appreciate the Reviewer's positive comments.

Comment: I mostly have some comments to the form of the manuscript.

Ln 52-54 This sentence is a strong statement, which is correct, but one or several references are needed.

Response: Per the Reviewer's suggestion, we have added multiple review references providing support for different portions of the statement in question (lines 52-54).

Comment: Line 107-115 Information on how the gene-modified mice got the GM is too sparse. Was FMT applied, was it by co-housing or???

Response: With apologies for the lack of clarity, we have added text to clarify that the GM of the genetically engineered mice (GEMs) was not manipulated in any way. Rather, samples were collected from GEMs immediately as they were being unpacked from shipping containers at our institution. Prior to shipping, GEMs were subjected to the diet and husbandry conditions of their respective home institution and facility (lines 160-163).

Comment: Ln 122 Why were the GM's termed high and low?

Response: Again, we regret the lack of clarity. The supplier-origin (SO) gut microbiomes (GMs) derived from mice purchased from the Jackson Laboratory and Inotiv (previously Envigo) were originally referred to as GM1 and GM4, and then renamed GM^{Low} and GM^{High}, due to the fact that those SO GMs had the lowest and highest mean richness. We have added text to clarify this (lines 123-124).

Comment: Ln 127-134 The information on the mice and their housing is very sparse. Which brand was the cages, bedding etc. Which cage type were included in the IVC's, what was the stocking density, which environmental enrichment was given, what was temperature, humidity etc. I know we are talking about a long period, but there must be some standards in the unit.

Response: We have added additional information related to animal husbandry in our facility (lines 129-138).

Comment: Also you cannot test for 'all known pathogens'. Write, which ones, eventually in a supplementary file, or just write the name of the IDEXX profile.

Response: We agree with the Reviewer's that 'all known pathogens' was an overstatement. We have added text to clarify the specific pathogens included in our quarterly and annual sentinel testing profiles (lines 139-152).

Comment: Ln 193-194 Which tests were used for normality and equality? Which software was used for statistics?

Response: With apologies for the omission, we have added text stating the methods used to test for normality (Shapiro-Wilk) and equal variance (Brown-Forsythe), and software used for statistical analyses (lines 218 and 222).

Comment: 'Results' It would make it easier to comprehend if subheading indicating the outcomes were used.

Response: We sincerely appreciate this suggestion and have added subheadings throughout the Results. We agree that these will help readers comprehend the outcomes in relation to each other.

Comment: 'Figure 1' The legend should be a bit more explanatory in relation to the meaning of intra- and inter. It is far easier to understand when reading the text in the manuscript.

Response: We appreciate the suggestion and have revised the legend for Figure 1 accordingly to improve clarity.

Comment: 'Figure 3' This could be two 3D-plots rather than four 2D-plots

Response: We considered this and explored 3D visualizations of these data. Doing so, however, made it difficult to fully show the variance captured by each of the first three coordinates and tended to make the distinction between certain groups (e.g., GEMs and GM3) less apparent. In our opinion, the use of separate panels with orthogonal views was the most effective way to demonstrate the variance among these coordinates. As such, we respectfully submit to retain Figure 3 as it stands.

Comment: Ln 289 *Helicobacter* in italics

Response: *Helicobacter* has been italicized here (line 324 in revised manuscript). We appreciate the Reviewer's attention to detail.

Comment: 'Figure 5A' Make the heat maps triangular to avoid double reporting. It makes it easier for the reader.

Response: We appreciate and agree with the Reviewer's suggestion. As such, both panels of Figure 5 have been revised.

Comment: Ln 330-332 This sentence is a statement, which should have one or several references, although it is correct.

Response: We agree completely and have added several references to support the sentence beginning on line 373 in the revised manuscript.

Reviewer #2:

Comment: The manuscript by Ericsson, et al. is well-written and timely for microbiome research. Research is often impacted by the microbiome present in genetically-engineered mice, especially in experimental reproducibility between laboratories, but has not been often carefully scrutinized until recent years. Describing the major effectors of microbiome composition is important for analysis of all animal research. The authors' prescription that the GM of GEM is an important characteristic to be recorded at the time of phenotyping is well supported and a critical one for both reproducibility and translation to human research. The data analysis is appropriate and thorough. While this paper adds substantially to the microbiome literature, there are several points that should be addressed before publication.

Response: We appreciate the Reviewer's positive comments.

Major points:

Comment: 1. "Laboratory environment" needs to be defined. Although the manuscript mentions diet in the introduction, no data is presented on the background of bedding/food/etc. for GEM lines submitted to the MU MMRRRC. Although this metadata may not be available, intra-lab similarities could be largely driven by diet decisions for each laboratory, which would also implicate the differences in intra-institution GMs. This type of conclusion is important for their discussion of implications of this work. Biosecurity in facilities may be easier or more difficult based on the source of microbes, and understanding action elements prescribed by this paper depend on which of these inputs is responsible for the variance. Presumably all of this is meant to be included under "laboratory environment" but what the authors intend to include within that catchall needs to be explicitly stated or more thoroughly discussed.

Response: The Reviewer makes an excellent suggestion. Accordingly, we have added text to the Discussion addressing these points. As some aspects of the immediate environment, e.g., biosecurity measures, are also relevant to the discussion of possible sources of *Helicobacter* in the GEM lines, the added text is also responsive to the issues raised below in reference to Figure S8 (lines 413-432).

Comment: a. Some of this may be able to be determined from local data- for instance MU lab C has split beta diversity based on whether a strain is B6/B6J or CJ_129 (Figure S2). Were those submissions split by time and or husbandry techniques as well, or can this be attributed mainly to mouse strain?

Response: We appreciate the Reviewer's suggestion and investigated each of the GEM lines clustering apart from the main cluster, including those mentioned by the Reviewer. As such, we have added a section in the Discussion acknowledging that, while the immediate environment represent a dominant effect on the microbiome of GEMs when viewed collectively, the genotype (including both genetic background and genetic modification) can also exert potent effects in individual GEMs (lines 433-452). The added text discusses the differences between those mice from MU lab C that cluster distinctly from the others, and another group of GEMs from a different institution that clusters distinctly from other GEMs from the same lab.

Comment: 2. The characteristics of institutions submitting animals to MU MMRRRC should be interrogated to determine if this dataset is representative of GEM lines in research facilities in general (i.e. is there a confounding variable associated with these institutions?). This could either be through a short analysis of publicly available data for *Helicobacter* species within mouse lines, or through analysis of published data. Setting this data within the larger community, given the lack of *Helicobacter* presence in SO GM is important. What other research settings lack *Helicobacter* in their colonies?

Response: The Reviewer makes an excellent suggestion. To provide more context regarding the institutions submitting GEM lines used in the current study and the generalizability of data from those lines, we have added text in the Discussion broadly describing the nature of the 84 different institutions (lines 453-457). Additionally, we have added a reference which performed a survey of publicly available metagenomic data from 1,091 SPF and 2,036 conventional laboratory mice (and 120 wild mice) for sequences annotated to specific enterohepatic *Helicobacter* spp. (Zhao et al. 2023, *Cell Reports*, 42, 112549). In that survey, *Helicobacter* spp. were largely absent from SPF mice but were detected in 24.9% of conventionally housed mice (still much lower than the prevalence in the current study). While that analysis incorporated a large number of samples ($N = 3,247$), they were drawn from a total of 54 studies; the current dataset represents 275 GEM lines. Drawing from older literature, a survey of 31 institutions

using PCR and RFLP analyses (Taylor et al. 2007, *J Clin Microbiol*, 45(7):2166) detected *Helicobacter* spp. in 84% of the samples. These and other references have been added to the Discussion to provide better context to our findings (lines 399-403, 425-428).

Comment: 3. Figure S8. The prevalence of *Helicobacter* found in these GEM lines is similar to what has been previously observed in laboratory mice in Europe, which correlated with prevalence in local wild mice (Wasimuddin, *Applied and Environmental Micro*, 2012). Because of the high importance the authors place on *Helicobacter* in GEM GM as compared to SO GM, possible sources of *Helicobacter* should be discussed.

Response: As mentioned above, the Discussion has been re-organized and expanded to address this and other Reviewer comments. Specifically, the prevalence of *Helicobacter* spp. in wild mice is addressed in lines 425-426 and lines 470-477.

Comment: 4. Figure S4. The interaction effect has a high negative value for F. This seems to indicate that the total variance is lower than that observed within the GM and/or strain values. The significant interaction effect should be better explained in the text in light of this value.

Response: The Reviewer raises a very interesting point. The authors included the interaction effect in Figure S4 for the sake of transparency and came to the same general conclusion as the Reviewer regarding the variance in the interaction relative to the variance in the main effects. That said, we intentionally neglected to mention it in the Results or Discussion for the following reasons. First, it is somewhat irrelevant to all of the main findings and conclusions of the current study. Second, it is not intuitive to readers familiar with traditional statistics wherein F values cannot be negative. Even for those familiar with multivariate statistics, a low *p* value *and* negative F value does not make sense. For these reasons, we felt that adding text discussing this detail in the analyses would unnecessarily confuse many readers without adding anything. If the Reviewer and/or Editor(s) feel that this point needs to be addressed in the manuscript, we are willing to make those edits. However, we respectfully request to leave the results in question in the supplementary figure for readers to interpret for themselves, but not to discuss it further in the text.

Minor points:

Comment: 1. No IACUC statement.

Response: We have added an IACUC statement including the IACUC protocol number (lines 164-166).

Comment: 2. Line 324- missing close parenthesis

Response: The close parenthesis has been added.

Comment: 3. Figure S3. No legend at right as stated in figure legend.

Response: With apologies for the omission, a legend has been added to Figure S3.

Comment: 4. Figure S3. Extra letter c behind vertical label in D.

Response: We appreciate the Reviewer's attention to detail. The errant letter c has been removed from the figure in question.

Comment: 5. Although the authors mention reproducibility multiple times, the other major reason that GM is important in animal experiments is translatability. This manuscript would be strengthened by a comparison to human microbiomes, as helicobacter is implicated not just in models but in human disease. Could the authors do a similar correlation to Figure 5 of helicobacter with important bacteroidota in publicly available human microbiome data? Although not necessary, it would increase the impact of this work.

Response: The Reviewer's suggestion is appreciated. While we believe that a survey of publicly available human data is beyond the scope of the current study, the issue of translatability merits discussion. Accordingly, we have added text to the Discussion addressing this topic in the context of the current data (lines 467-477).

Re: mSystems01112-25R1 (Dominant effects of the immediate environment on the gut microbiome of mice used in biomedical research)

Dear Dr. Aaron Conrad Ericsson:

I have closely read your manuscript to ensure that the reviewer comments were addressed, and consider it very near to publication.

The one response that I question is to Review #1's last comment regarding L 330-332 in the original manuscript, to which you report added references beginning at L. 373. Neither line number seemed to match either the original or revised manuscript exactly, but I don't see any references added to this section of the manuscript at all (or at least these aren't indicated in the markup). Please check this and clarify.

I have three other very minor comments:

- (1) LEfSe and SourceTracker versions should be added.
- (2) Genus names in Figure 5 should be italicized.
- (3) In Figure S3A, the letter indicating the ANOVA result is missing for the left-most group (GM4).

Revision Guidelines

Sincerely,

Jonathan Klassen
Editor
mSystems

The authors sincerely appreciate the initial reviews and the careful reading and editorial review of our revised text. Below are our responses to each of those comments resulting from editorial review.

Comment: The one response that I question is to Review #1's last comment regarding L 330-332 in the original manuscript, to which you report added references beginning at L. 373. Neither line number seemed to match either the original or revised manuscript exactly, but I don't see any references added to this section of the manuscript at all (or at least these aren't indicated in the markup). Please check this and clarify.

Response: We apologize for the confusion as it appears that the 'track changes' function was not active for those edits. We have verified that the sentence in question (originally beginning on Line 330) now begins on line 373 in the revised manuscript, and several references were indeed added to support the statement.

Comment: I have three other very minor comments:
(1) LEfSe and SourceTracker versions should be added.

Response: We have added the version numbers for these tools (lines 224 and 225).

Comment: (2) Genus names in Figure 5 should be italicized.

Response: We appreciate the Editor's attention to detail. Genus names have been italicized in the revised version of Figure 5.

Comment: (3) In Figure S3A, the letter indicating the ANOVA result is missing for the left-most group (GM4).

Response: There is no letter over that group due to the fact that none of the pairwise comparisons including that group yielded $p < 0.05$. This was unexpected (mentioned on line 269) and may reflect the low sample size in that genetic background.

The following pages contain our initial Reviewer comments and responses.

The authors greatly appreciate the Reviewers' careful reading of our manuscript and their thoughtful suggestions. We address each of the Reviewer's comments individually below.

Reviewer #1:

Comment: This manuscript by Ericsson et al is relevant, and the setup with four GM's and all these gene modified mice is unique. The result is not that surprising, but it always important to have documentation for our assumptions. As underlined in the discussion reproducibility of mouse studies would need much more consideration of the GM.

Response: We appreciate the Reviewer's positive comments.

Comment: I mostly have some comments to the form of the manuscript.

Ln 52-54 This sentence is a strong statement, which is correct, but one or several references are needed.

Response: Per the Reviewer's suggestion, we have added multiple review references providing support for different portions of the statement in question (lines 52-54).

Comment: Line 107-115 Information on how the gene-modified mice got the GM is too sparse. Was FMT applied, was it by co-housing or???

Response: With apologies for the lack of clarity, we have added text to clarify that the GM of the genetically engineered mice (GEMs) was not manipulated in any way. Rather, samples were collected from GEMs immediately as they were being unpacked from shipping containers at our institution. Prior to shipping, GEMs were subjected to the diet and husbandry conditions of their respective home institution and facility (lines 160-163).

Comment: Ln 122 Why were the GM's termed high and low?

Response: Again, we regret the lack of clarity. The supplier-origin (SO) gut microbiomes (GMs) derived from mice purchased from the Jackson Laboratory and Inotiv (previously Envigo) were originally referred to as GM1 and GM4, and then renamed GM^{Low} and GM^{High}, due to the fact that those SO GMs had the lowest and highest mean richness. We have added text to clarify this (lines 123-124).

Comment: Ln 127-134 The information on the mice and their housing is very sparse. Which brand was the cages, bedding etc. Which cage type were included in the IVC's, what was the stocking density, which environmental enrichment was given, what was

temperature, humidity etc. I know we are talking about a long period, but there must be some standards in the unit.

Response: We have added additional information related to animal husbandry in our facility (lines 129-138).

Comment: Also you cannot test for 'all known pathogens'. Write, which ones, eventually in a supplementary file, or just write the name of the IDEXX profile.

Response: We agree with the Reviewer's that 'all known pathogens' was an overstatement. We have added text to clarify the specific pathogens included in our quarterly and annual sentinel testing profiles (lines 139-152).

Comment: Ln 193-194 Which tests were used for normality and equality? Which software was used for statistics?

Response: With apologies for the omission, we have added text stating the methods used to test for normality (Shapiro-Wilk) and equal variance (Brown-Forsythe), and software used for statistical analyses (lines 218 and 222).

Comment: 'Results' It would make it easier to comprehend if subheading indicating the outcomes were used.

Response: We sincerely appreciate this suggestion and have added subheadings throughout the Results. We agree that these will help readers comprehend the outcomes in relation to each other.

Comment: 'Figure 1' The legend should be a bit more explanatory in relation to the meaning of intra- and inter. It is far easier to understand when reading the text in the manuscript.

Response: We appreciate the suggestion and have revised the legend for Figure 1 accordingly to improve clarity.

Comment: 'Figure 3' This could be two 3D-plots rather than four 2D-plots

Response: We considered this and explored 3D visualizations of these data. Doing so, however, made it difficult to fully show the variance captured by each of the first three coordinates and tended to make the distinction between certain groups (e.g., GEMs and GM3) less apparent. In our opinion, the use of separate panels with orthogonal views was the most effective way to demonstrate the variance among these coordinates. As such, we respectfully submit to retain Figure 3 as it stands.

Comment: Ln 289 *Helicobacter* in italics

Response: *Helicobacter* has been italicized here (line 324 in revised manuscript). We appreciate the Reviewer's attention to detail.

Comment: 'Figure 5A' Make the heat maps triangular to avoid double reporting. It makes it easier for the reader.

Response: We appreciate and agree with the Reviewer's suggestion. As such, both panels of Figure 5 have been revised.

Comment: Ln 330-332 This sentence is a statement, which should have one or several references, although it is correct.

Response: We agree completely and have added several references to support the sentence beginning on line 373 in the revised manuscript.

Reviewer #2:

Comment: The manuscript by Ericsson, et al. is well-written and timely for microbiome research. Research is often impacted by the microbiome present in genetically-engineered mice, especially in experimental reproducibility between laboratories, but has not been often carefully scrutinized until recent years. Describing the major effectors of microbiome composition is important for analysis of all animal research. The authors' prescription that the GM of GEM is an important characteristic to be recorded at the time of phenotyping is well supported and a critical one for both reproducibility and translation to human research. The data analysis is appropriate and thorough. While this paper adds substantially to the microbiome literature, there are several points that should be addressed before publication.

Response: We appreciate the Reviewer's positive comments.

Major points:

Comment: 1. "Laboratory environment" needs to be defined. Although the manuscript mentions diet in the introduction, no data is presented on the background of bedding/food/etc. for GEM lines submitted to the MU MMRRC. Although this metadata may not be available, intra-lab similarities could be largely driven by diet decisions for each laboratory, which would also implicate the differences in intra-institution GMs. This type of conclusion is important for their discussion of implications of this work. Biosecurity in facilities may be easier or more difficult based on the source of microbes, and understanding action elements prescribed by this paper depend on which of these inputs is responsible for the variance. Presumably all of this is meant to be included

under "laboratory environment" but what the authors intend to include within that catchall needs to be explicitly stated or more thoroughly discussed.

Response: The Reviewer makes an excellent suggestion. Accordingly, we have added text to the Discussion addressing these points. As some aspects of the immediate environment, e.g., biosecurity measures, are also relevant to the discussion of possible sources of *Helicobacter* in the GEM lines, the added text is also responsive to the issues raised below in reference to Figure S8 (lines 413-432).

Comment: a. Some of this may be able to be determined from local data- for instance MU lab C has split beta diversity based on whether a strain is B6/B6J or CJ_129 (Figure S2). Were those submissions split by time and or husbandry techniques as well, or can this be attributed mainly to mouse strain?

Response: We appreciate the Reviewer's suggestion and investigated each of the GEM lines clustering apart from the main cluster, including those mentioned by the Reviewer. As such, we have added a section in the Discussion acknowledging that, while the immediate environment represent a dominant effect on the microbiome of GEMs when viewed collectively, the genotype (including both genetic background and genetic modification) can also exert potent effects in individual GEMs (lines 433-452). The added text discusses the differences between those mice from MU lab C that cluster distinctly from the others, and another group of GEMs from a different institution that clusters distinctly from other GEMs from the same lab.

Comment: 2. The characteristics of institutions submitting animals to MU MMRRC should be interrogated to determine if this dataset is representative of GEM lines in research facilities in general (i.e. is there a confounding variable associated with these institutions?). This could either be through a short analysis of publicly available data for *Helicobacter* species within mouse lines, or through analysis of published data. Setting this data within the larger community, given the lack of *Helicobacter* presence in SO GM is important. What other research settings lack *Helicobacter* in their colonies?

Response: The Reviewer makes an excellent suggestion. To provide more context regarding the institutions submitting GEM lines used in the current study and the generalizability of data from those lines, we have added text in the Discussion broadly describing the nature of the 84 different institutions (lines 453-457). Additionally, we have added a reference which performed a survey of publicly available metagenomic data from 1,091 SPF and 2,036 conventional laboratory mice (and 120 wild mice) for sequences annotated to specific enterohepatic *Helicobacter* spp. (Zhao et al. 2023, *Cell Reports*, 42, 112549). In that survey, *Helicobacter* spp. were largely absent from SPF mice but were detected in 24.9% of conventionally housed mice (still much lower than the prevalence in the current study). While that analysis incorporated a large number of

samples ($N = 3,247$), they were drawn from a total of 54 studies; the current dataset represents 275 GEM lines. Drawing from older literature, a survey of 31 institutions using PCR and RFLP analyses (Taylor et al. 2007, *J Clin Microbiol*, 45(7):2166) detected *Helicobacter* spp. in 84% of the samples. These and other references have been added to the Discussion to provide better context to our findings (lines 399-403, 425-428).

Comment: 3. Figure S8. The prevalence of *Helicobacter* found in these GEM lines is similar to what has been previously observed in laboratory mice in Europe, which correlated with prevalence in local wild mice (Wasimuddin, *Applied and Environmental Micro*, 2012). Because of the high importance the authors place on *Helicobacter* in GEM GM as compared to SO GM, possible sources of *Helicobacter* should be discussed.

Response: As mentioned above, the Discussion has been re-organized and expanded to address this and other Reviewer comments. Specifically, the prevalence of *Helicobacter* spp. in wild mice is addressed in lines 425-426 and lines 470-477.

Comment: 4. Figure S4. The interaction effect has a high negative value for F. This seems to indicate that the total variance is lower than that observed within the GM and/or strain values. The significant interaction effect should be better explained in the text in light of this value.

Response: The Reviewer raises a very interesting point. The authors included the interaction effect in Figure S4 for the sake of transparency and came to the same general conclusion as the Reviewer regarding the variance in the interaction relative to the variance in the main effects. That said, we intentionally neglected to mention it in the Results or Discussion for the following reasons. First, it is somewhat irrelevant to all of the main findings and conclusions of the current study. Second, it is not intuitive to readers familiar with traditional statistics wherein F values cannot be negative. Even for those familiar with multivariate statistics, a low p value *and* negative F value does not make sense. For these reasons, we felt that adding text discussing this detail in the analyses would unnecessarily confuse many readers without adding anything. If the Reviewer and/or Editor(s) feel that this point needs to be addressed in the manuscript, we are willing to make those edits. However, we respectfully request to leave the results in question in the supplementary figure for readers to interpret for themselves, but not to discuss it further in the text.

Minor points:

Comment: 1. No IACUC statement.

Response: We have added an IACUC statement including the IACUC protocol number (lines 164-166).

Comment: 2. Line 324- missing close parenthesis

Response: The close parenthesis has been added.

Comment: 3. Figure S3. No legend at right as stated in figure legend.

Response: With apologies for the omission, a legend has been added to Figure S3.

Comment: 4. Figure S3. Extra letter c behind vertical label in D.

Response: We appreciate the Reviewer's attention to detail. The errant letter c has been removed from the figure in question.

Comment: 5. Although the authors mention reproducibility multiple times, the other major reason that GM is important in animal experiments is translatability. This manuscript would be strengthened by a comparison to human microbiomes, as helicobacter is implicated not just in models but in human disease. Could the authors do a similar correlation to Figure 5 of helicobacter with important bacteroidota in publicly available human microbiome data? Although not necessary, it would increase the impact of this work.

Response: The Reviewer's suggestion is appreciated. While we believe that a survey of publicly available human data is beyond the scope of the current study, the issue of translatability merits discussion. Accordingly, we have added text to the Discussion addressing this topic in the context of the current data (lines 467-477).

Re: mSystems01112-25R2 (Dominant effects of the immediate environment on the gut microbiome of mice used in biomedical research)

Dear Dr. Aaron Conrad Ericsson:

Your manuscript has been accepted, and I am forwarding it to the ASM production staff for publication. Your paper will first be checked to make sure all elements meet the technical requirements. ASM staff will contact you if anything needs to be revised before copyediting and production can begin. Otherwise, you will be notified when your proofs are ready to be viewed.

Sincerely,
Jonathan Klassen
Editor
mSystems